Comprehensive phenotyping and transcriptome profiling to study nanotoxicity in C. elegans

Viau Charles 1
Haçariz Orçun 1
Karimian Farial 1
Xia Jianguo jeff.xia@mcgill.ca 1 2
1 Institute of Parasitology, McGill University , Montreal , Canada
2 Department of Animal Science, McGill University , Montreal , Quebec , Canada
Mortimer Monika
Electronic publication date: 2020 Feb 27
Publication date: 2020
Volume: 8
Electronic Location ID: e8684
Received 2019 Oct 10; Accepted 2020 Feb 4
Copyright: ©2020 Viau et al.
Copyright year: 2020
Copyright holder: Viau et al.
License: This is an open access article distributed under the terms of the Creative Commons Attribution License, which permits unrestricted use, distribution, reproduction and adaptation in any medium and for any purpose provided that it is properly attributed. For attribution, the original author(s), title, publication source (PeerJ) and either DOI or URL of the article must be cited.
License URL: https://creativecommons.org/licenses/by/4.0/

Keywords: C. elegans, Ag, SiO2, Nanoparticles, Locomotion Velocity, Growth Inhibition, Reproduction, Neurotoxicity, RNAseq

Funding: Fonds Québécois de la Recherche sur la Nature et les Technologies (FQRNT), Genome Canada, Genome Quebec The Natural Sciences and Engineering Research Council of Canada (NSERC) Discovery Grant This study was supported by the Fonds Québécois de la Recherche sur la Nature et les Technologies (FQRNT), Genome Canada, Genome Quebec and the Natural Sciences and Engineering Research Council of Canada (NSERC) Discovery Grant. The funders had no role in study design, data collection and analysis, decision to publish, or preparation of the manuscript.

==============================
Engineered nanoparticles are used at an increasing rate in both industry and medicine without fully understanding their impact on health and environment. The nematode Caenorhabditis elegans is a suitable model to study the toxic effects of nanoparticles as it is amenable to comprehensive phenotyping, such as locomotion, growth, neurotoxicity and reproduction. In this study, we systematically evaluated the effects of silver (Ag) and five metal oxide nanoparticles: SiO2, CeO2, CuO, Al2O3 and TiO2. The results showed that Ag and SiO2 exposures had the most toxic effects on locomotion velocity, growth and reproduction, whereas CeO2, Al2O3 and CuO exposures were mostly neurotoxic. We further performed RNAseq to compare the gene expression profiles underlying Ag and SiO2toxicities. Gene set enrichment analyses revealed that exposures to Ag and SiO2consistently downregulated several biological processes (regulations in locomotion, reproductive process and cell growth) and pathways (neuroactive ligand-receptor interaction, wnt and MAPK signaling, etc.), with opposite effects on genes involved in innate immunity. Our results contribute to mechanistic insights into toxicity of Ag and SiO2 nanoparticles and demonstrated that C. elegans as a valuable model for nanotoxicity assessment.

Introduction

The use of engineered nanoparticles has increased enormously over the last decade, and the nanotechnology industry has grown from a 10-billion-dollar enterprise in 2012 to being valued over one trillion dollars in 2015 (Gao et al., 2011). However, the potential impacts of these nanoparticles on environment and animals have not been fully characterized and further research is warranted. Nanoparticles are defined as particulate matter ranging from 1 to 100 nm in size with properties not exhibited by their larger bulk counterparts (Khanna et al., 2015; Capco & Chen, 2014; Maynard, 2011; Djurišić et al., 2015). The reactivity of nanoparticles depends on their size, charge, dose as well as the chemical composition of their coating (Medina et al., 2007). For instance, the surface area of smaller nanoparticles is larger compared to their larger counterparts, meaning they are more reactive and hence have a larger propensity of being toxic (Oberdörster, 2010).

Caenorhabditis elegans is a free-living soil nematode reaching approximately one millimeter in length in adult stage and has a relatively simple life cycle that can be grown on solid (i.e., nematode growth medium, NGM) or liquid media (i.e., S-medium) (Brenner, 1974; Lewis & Fleming, 1995). The small size and relatively cheap maintenance cycle of C. elegans make the nematode very amenable for various phenotype screening. C. elegans has been well-established as an in vivo model for testing the effects of heavy metals and novel anthelminthics (Kaletta & Hengartner, 2006; Ruiz-Lancheros et al., 2011). In terms of conservation of genes and biological pathways with humans, C. elegans shares 60 to 80% gene homology and possesses 12 of the 17 known signal transduction pathways (Kaletta & Hengartner, 2006; National Research Council, 2000).

To validate our use of C. elegans as a model for nanotoxicity, as this bacterivore worm constantly interacts with microbes in nature, which are ingested through the pharynx, the main potential route of exposure to nanoparticles is consequently oral (Pluskota et al., 2009). Similarly, human exposure to nanoparticles is also mostly through an oral route of entry, as nanoparticles are added to food in significant amount; the most prevalent ones being Ag, SiO2, TiO2 and ZnO (Fröhlich & Roblegg, 2016). For example, it is estimated a 70 kg individual ingests 126 mg of Ag nanoparticles per day in Europe (Dekkers et al., 2011). The nematode worm C. elegans, as a model organism, is thus valid for studying nanotoxicity in higher eukaryotic organisms such as humans. Additionally, a second route of exposure is through the worm’s vulval slit, where nanoparticles interfere with vulval cells and spermatecae (Scharf, Gührs & Von Mikecz, 2016). However, two routes of exposure to nanoparticles that cannot be studied in C. elegans are respiratory and dermal absorption, which are prevalent routes of exposure to nanoparticles for humans (Fröhlich & Roblegg, 2016).

The fast phenotyping of C. elegans can be coupled with transcriptome profiling (i.e., gene expression microarray or RNAseq) to study underlying molecular mechanisms. For instance, using gene expression microarray, Rocheleau et al. found that C. elegans exposed to nano-TiO2 showed increased expression of the glutathione S-transferase gene gst-3 and the cytochrome P450 gene cyp33-11; while the oxidative stress response, as measured by the stress resistance regulator scl-1, showed increased expression after exposure to both nano- and bulk-sized TiO2 (Rocheleau et al., 2015). In addition, the expression of pod-2, a reproduction-related gene, was decreased in a concentration-dependent manner with nano-TiO2 exposure (Rocheleau et al., 2015). Based on the same technology, Starnes et al. identified that five lysosomal pathway related genes, including genes encoding the cysteine proteases cpr-1 and cpr-2, were changed significantly after exposure to silver (Ag) nanoparticles (Starnes et al., 2016). To the best of our knowledge, no transcriptome profiling has been reported to investigate SiO2 nanoparticles in C. elegans. The main objective of the current study is to develop and to evaluate a C. elegans-based animal model to study nanotoxicity by integrating comprehensive phenotyping and transcriptome profiling. We selected Ag and five metal oxide nanoparticles (SiO2, TiO2, CuO, Al2O3 and CeO2), and measured four endpoints (locomotion velocity, growth, reproduction and neurotoxicity) in C. elegans after exposure to these nanoparticles. Worms that exhibited the most significant effects were subjected to RNAseq to identify the affected biological processes and pathways. Hence, we offer a novel perspective to study nanoparticle toxicity using the soil nematode C. elegans.

Material and Methods

Caenorhabditis elegans culture

The C. elegans N2 strain was obtained from the Caenorhabditis Genetics Center (CGC) at the University of Minnesota. Escherichia coli OP50 was also obtained from the CGC and was grown for 18 h at 37 °C in Luria-Bertani (LB) broth (Bertani, 1951). The N2 strain was maintained at 21 °C in an incubator on Nematode Growth Media (NGM) plates and C. elegans were synchronized using 5.0 ml of alkaline bleach to kill the adult hermaphrodites and release their eggs (Stiernagle, 2006). Eggs were then washed three times with M9 buffer and left overnight on a rocking platform at room temperature to hatch into L1 larvae (Stiernagle, 2006).

Preparation of nanoparticles

Ag, SiO2, CuO, Al2O3, and CeO2 nanoparticles were purchased from Sigma-Aldrich (St. Louis, USA). TiO2 nanoparticles were obtained from the Joint Research Center of (JRC) the European Commission. All nanoparticles were less than 100 nm in size as described by the manufacturer and commission. Product details are shown in Table 1. Nanoparticles were dissolved at stock concentration of 1,000 µg/ml in ddH2O and sonicated using an Ultrasonic Processor VCX (GEX) 750 at an amplitude of 40% for a 3-minute pulse, followed by 1 min on ice. This step was repeated five times to ensure complete disaggregation of the nanoparticles. Nanoparticle solutions were then diluted to working concentrations of 200 µg/ml in S-medium (Jung et al., 2015).

Locomotion velocity and growth (body length) assays

200 L1 stage C. elegans N2, obtained after synchronization with alkaline bleach, were grown in S-medium in 6-well plates containing 0 (control), 10 or 50 µg/ml of each nanoparticle, supplemented with E. coli OP50 at a final optical density at 595 nm (OD595) of 1, for 72 h at 21 °C until they reached the day 1 adult stage. For the locomotion velocity endpoint assay, worms were then washed once in 1X M9 buffer and placed on unseeded NGM plates and allowed to explore their surroundings for 10 min. Worms (n = 14 to 111 per condition) were then recorded using a Nikon camera (SMZ1270) linked to a computer. The average locomotion velocity of each worm was calculated for 30 s at an interval of 0.500 ms using the software (NIS-Elements, version 4.60) accompanying the camera. The average locomotion velocity was calculated by averaging the locomotion velocity (in µm/s) over the 30 s of recording. For the growth (body length) endpoint assay, worms were grown in the same manner as in the locomotion velocity endpoint assay, washed once in 1X M9 buffer and killed with 10 mM sodium azide. Dead worms were transferred to an unseeded NGM plate to take pictures. The body length of worms (n = 20 to 43 per condition), measured in µm, was calculated using the camera’s software (NIS-Elements, version 4.60).

Reproduction (brood size) assay

200 L1 C. elegans N2, obtained by synchronization, were grown for 48 h at 21 °C to the L4-young adult stage on E. coli OP50-seeded NGM plates. Five L4-young adult hermaphrodites were transferred to an individual well in quadruplicate of a 12-well plate containing S-medium supplemented with either 0 (control), 10 or 50 µg/ml of each nanoparticle and E. coli OP50. The L4-young-adult hermaphrodites were then allowed to grow and lay eggs for 96 h, and resulting progeny were counted by dilution.

Neurotoxicity (number of head thrashes) assay

200 L1 C. elegans N2 were grown to the adult day 1 stage (72 h at 21 °C) in individual wells of a 6-well plate containing S-medium containing either 0 (control), 10 or 50 µg/ml of each nanoparticle and E. coli OP50. A total of 1.0 ml of the well contents were centrifuged at 1,000 rpm for 2 min, the supernatant was decanted, leaving the worm pellet undisturbed. Worms were then washed in 1X M9 buffer and centrifuged at 1,000 rpm for 2 min. Worms were then transferred to an unseeded NGM plate containing 60 µl of K-medium (2.36 g of KCl and 3.0 g NaCl per liter of media dissolved in ddH2O). Individual adult day 1 stage C. elegans were transferred into the drop of K-medium. Worms were allowed to swim freely for 1 min. Afterwards, the number of head thrashes of each individual worm (n = 8 to 55 per condition) were counted for 1 min as described by Tsalik & Hobert (2003).

Table 1 Product details of six different nanoparticles.

Each nanoparticle, used in this study, is smaller than 100 nm in size. NA: non-applicable (no information provided from the suppliers).

Nanoparticle	Symbol	Catalog number/Brand	CAS number	Size (nm)	Shape	
Silver	Ag	576832/Aldrich	7440-22-4	<100	Spherical	
Silicon dioxide	SiO2	637238/Aldrich	7631-86-9	10-20 (BET)	Spherical	
Cerium(IV) oxide	CeO2	544841/Aldrich	1306-38-3	<25 (BET)	NA	
Copper(II) oxide	CuO	544868/Aldrich	1317-38-0	<50 (TEM)	NA	
Aluminum oxide	Al2O3	544833/Aldrich	1344-28-1	<50 (TEM)	NA	
Titanium dioxide	TiO2	NM-101/JRC	NA	8	NA	

Total RNA extraction of worms exposed to Ag and SiO2 nanoparticles

400 L1 C. elegans N2 were grown in individual wells of a 12-well plate containing either 0, 10 µg/ml Ag nanoparticles or 10 µg/ml SiO2 nanoparticles and supplemented with E. coli OP50, until the worms reached the adult day 1 stage (72 h at 21 °C). Each condition was repeated six times. The contents of the wells were centrifuged at 1,000 rpm for 2 min, the supernatant was decanted and the worm pellet was washed twice with 1X M9 buffer. 200 µl of Trizol (Ambion, USA) was then added to the worm pellet. The worm pellet then was flash-frozen in liquid nitrogen, followed by a quick thaw. These two steps were repeated once. RNA from the resulting worm pellet-Trizol solution was extracted using the Direct-zol RNA miniprep kit (Zymo Research, USA) according to the manufacturer’s instructions. Quantity and purity of total RNA were checked using a spectrophotometer (ND-1000, NanoDrop). The RNA samples were then sent to the McGill University and Génome Québec Innovation Centre (http://gqinnovationcenter.com) for quality analysis with Bioanalyser and for single-end read (100 base) RNA sequencing under HiSeq 2500 (Illumina).

Data analysis for RNAseq

Raw data for each sample was received in fastq file format from the McGill University and Génome Québec Innovation Centre. Read quality was checked with FASTQC (version 0.11.3) and adapter related sequences were removed using Trim Galore (version 0.4.5) (https://www.bioinformatics.babraham.ac.uk/projects/). The genome sequence of C. elegans and GTF file (Caenorhabditis_elegans.WBcel235.91.gtf) were downloaded from ENSEMBL (https://www.ensembl.org/). Reads were aligned to the C. elegans genome with HISAT2 (version 2.1.0) (Kim, Langmead & Salzberg, 2015) and sorted alignment files were generated by SAMtools (version 1.7) (Li et al., 2009). Raw read counts were extracted using HTSeq (version 0.9.1) with the intersection-strict mode (Anders, Pyl & Huber, 2015). Entrez IDs were extracted from a Bioconductor package (org.Ce.eg.db) (Carlson, 2018) and assigned to the wormbase gene sequences using R. Sample distribution by principal component analysis was visualised using NetworkAnalyst 3.0 (Zhou et al., 2019). Differential gene expression analysis between the nanoparticle treatments and control was carried out using edgeR where data were normalised by trimmed mean of M-values (TMM) and tag-wise dispersion parameters were estimated using the empirical Bayes method (Robinson, McCarthy & Smyth, 2010). For gene set enrichment analysis, genes were ranked by the expression ratio (combination of log2 fold change and FDR) and normalized enrichment score (NES) was determined using GSEAPreranked in Gene Set Enrichment Analysis (GSEA; version 3.0) (Subramanian et al., 2005). The value for the parameter of min size: exclude smaller sets was set to 0, the value for permutations was set to 1000 and the enrichment statistic was set to classic. For use with GSEAPreranked, GO derived MSigDB format gene sets for C. elegans was downloaded from GO2MSIG (Powell, 2014) and KEGG database of C. elegans was extracted from a current Bioconductor package (version 3.7) (Luo et al., 2009) and converted to *gmt file. Pathway interaction was investigated using ClueGO (Bindea et al., 2009). Furthermore, gene enrichment in GO and newly determined terms were carried out using GOATOOLS (Klopfenstein et al., 2018) and WormExp (Yang, Dierking & Schulenburg, 2016), respectively. Differentially expressed genes in the toxicity groups (compared to control) were further searched to ascertain whether they were reported in metal toxicity based on previous studies (Caito et al., 2012; Cui et al., 2007; Roh, Lee & Choi, 2006; Kumar et al., 2015; Anbalagan et al., 2012).

Statistical analysis

Statistical analysis was performed using GraphPad Prism (version 8.0.0). The statistical difference between the groups in the toxicity assays was evaluated with One-way ANOVA followed by Dunnett’s multiple comparison test and P value less than 0.05 was accepted as statistically significant.

Results

The pooled average locomotion velocity for C. elegans grown under control conditions was 150.1 µm/s across all treatments, indicating that worms are active after growth in S-medium for 72 h at 21 °C. In comparison, the average locomotion velocity of worms decreased to 91.5 µm/s in the presence of 10 µg/ml Ag nanoparticles and further reduced to 44.2 µm/s in the presence of 50 µg/ml Ag nanoparticles, which was statistically significant for each concentration (P = 0.0001 and P < 0.0001, respectively) (Fig. 1A). Additionally, the difference in the reduction levels by the concentrations (>50%) indicated a dose-dependent decrease in average locomotion velocity in response to increasing Ag nanoparticle concentration. The second nanoparticle to influence the worms’ average locomotion velocity was SiO2, which caused significant decreases at 10 or 50 µg/ml doses in comparison with control (P < 0.0001 for both concentrations). Surprisingly, it appeared that TiO2 nanoparticles increased C. elegans average locomotion velocity at 10 µg/ml (P = 0.0257), although this could be a statistical artifact due to sampling effect based on the data distribution (Fig. 1C). Al2O3-,TiO2- and CuO-treated C. elegans showed decreases in average locomotion velocity when tested at 50 µg/ml (P = 0.0196, for Al2O3 and P < 0.005 for both TiO2 and CuO), displaying locomotion velocities of 104.4, 114.8 and 112.2 µm/s, respectively (Figs. 1C–1E). CeO2 nanoparticles had no effect at any of the concentrations tested (P > 0.05, for both concentrations).

Figure 1 Average locomotion velocity of adult day 1 C. elegans N2 after exposure to various nanoparticles (A, Ag; B, SiO2; C, TiO2; D, CeO2; E, Al2O3; and F, CuO) at 0, 10 and 50 µg/ml.

A total of 200 L1s were grown in 6-well plates containing S-medium supplemented with described concentrations of each nanoparticle for 72 h at 21 °C. After washing and transferring the worms onto an unseeded NGM plate, adult day 1 C. elegans were then video-recorded for 30 s using a Nikon camera and average locomotion velocity was calculated using corresponding software for every 0.500 msec over the 30 s time span. Only Ag and SiO2 nanoparticles show significant reductions in the velocity parameter at 10 µg/ml in comparison with control (P < 0.0001). Each point represents a single worm. Statistical difference is indicated with an asterisk (*) (*P < 0.05, ** P < 0.01, *** P = 0.0001, **** P < 0.0001 and ns: non-significant).

For the growth inhibition (body length) assay, Ag nanoparticles had the greatest effect. The pooled average body length of worms grown in the control S-medium after a three-day incubation at 21 °C was 1199.6 µm, indicating that the worms grew efficiently in this medium. As with the locomotion velocity endpoint assay, a concentration-dependent decrease in body length was observed when worms were exposed to different concentrations of Ag nanoparticles, with average body lengths of 1017.8 µm under 10 µg/ml and 859.3 µm under 50 µg/ml, respectively (P < 0.0001 for both concentrations) (Fig. 2A). Exposure to SiO2 nanoparticles induced significant decrease at 10 µg/ml (P < 0.0001) (Fig. 2B), but no further significant decrease was observed at 50 µg/ml. Exposure to CuO nanoparticles also resulted in a concentration-dependent decrease in body length, leading to an average body length of 1055.3 and 923.1 µm, respectively (P < 0.0001 for both concentrations) (Fig. 2F). For CeO2 and Al2O3 nanoparticles, significant decreases in body length were observed at 10 µg/ml compared to control worms (P < 0.0001, Figs. 2D and 2E). For TiO2 nanoparticles, a significant difference was observed only at 50 µg/ml, in which the exposed worms showed an average body length of 1130.8 µm (P = 0.001) (Fig. 2C). In summary, these results show Ag and SiO2 nanoparticles have similar toxicity on C. elegans, although the effect appears to be concentration-dependent for Ag nanoparticles whereas there is likely a threshold effect for SiO2 nanoparticles.

Figure 2 Body length of adult day 1 C. elegans. N2 after exposure to various nanoparticles (A, Ag; B, SiO2; C, TiO2; D, CeO2; E, Al2O3; and F, CuO) at 0, 10 and 50 µg/ml.

200 L1s were grown in 6-well plates containing S-medium supplemented with described concentrations of each nanoparticle for 72 h at 21 °C. Worms were killed using 10 mM sodium azide and transferred onto an unseeded NGM plate. Worms were photographed using a Nikon camera and body length was determined using corresponding software. All the nanoparticles except TiO2 demonstrated reductions in the body length at 10 µg/ml, compared to control (P < 0.0001). Each point represents a single worm.

To measure the effects of the various nanoparticles on C. elegans reproduction, we incubated five L4-young adults in S-medium supplemented with E. coli OP50 and the respective nanoparticles for four days (96 h) at 21 °C. It was expected that each worm would lay approximately 300 eggs in that time span, so that the total number of progeny per 5 worms would be near 1,500 under control conditions (Sulston & Hodgkin, 1988; Sonowal et al., 2017). The average control value of progeny produced by 5 worms under our experimental conditions was 1,288 after 96 h. We found that most nanoparticles reduced the number of progeny produced by C. elegans. At 10 µg/ml, Ag nanoparticles decreased the brood size of C. elegans to around 37%, which was statistically significant (P < 0.0001) (Fig. 3A). The effect was even more pronounced at 50 µg/ml, as Ag nanoparticles decreased the number of progeny to 33% of the control value, suggesting that these nanoparticles do indeed decrease C. elegans brood size (P < 0.0001) (Fig. 3A). On the other hand, SiO2 nanoparticles decreased brood size substantially at both 10 µg/ml and 50 µg/ml (P < 0.0001) (Fig. 3B), indicating that SiO2 nanoparticles are potent inhibitors of C. elegans reproduction in concentrations ranging in µg/ml. In contrast, TiO2 nanoparticles, known to inhibit C. elegans reproduction, reduced brood size to about 80% of the control value in our testing concentration range (Fig. 3C). CeO2 nanoparticles inhibited C. elegans reproduction as well. The brood size decreased to 55% of the control-treated value at a concentration of 10 µg/ml (P = 0.0002) (Fig. 3D). Interestingly, at 50 µg/ml, the decrease was not as pronounced, equating to 89% of the control-treated value (P > 0.05) (Fig. 3D). This observation may be due to the aggregation of CeO2 at higher concentrations. Al2O3 nanoparticles did not show statistically significant effects on C. elegans under our conditions (P > 0.05) (Fig. 3E), whereas CuO nanoparticles decreased the brood size value to 83% and 71% of the control value, at 10 µg/ml and 50 µg/ml, respectively (Fig. 3F).

Figure 3 Reproduction capacity of C. elegans N2 after exposure to various nanoparticles (A, Ag; B, SiO2; C, TiO2; D, CeO2; E, Al2O3; and F, CuO) at 0, 10 and 50 µg/ml.

200 L1s obtained from synchronization were seeded onto E. coli OP50-coated NGM plates and grown to the L4-young adult stage (48 h at 21 °C). Five L4-young adults were then transferred to a single well of a 12-well plate containing S-medium with corresponding concentrations of nanoparticles (0,10 and 50 µg/ml). Plates were then incubated for 96 h at 21 °C. The resulting total number of progeny was then calculated by dilution. Ag, SiO2 and CeO2 nanoparticles reduced the reproduction capacity in comparison with control significantly at 10 µg/ml (P < 0.0001). Bar-graphs represent average brood size ± standard deviation (SD) per five L4-young adult nematodes per condition.

Both Ag and SiO2 nanoparticles showed no significant impact of the number of head thrashes based on a population of worms in 60 s at 10 µg/ml versus control worms (Figs. 4A and 4B). SiO2 nanoparticles showed a slight effect on the neurotoxicity assay at 50 µg/ml (P = 0.0033), whereas Ag nanoparticles had no such effect on neurotoxicity. Similar results were observed for TiO2 and nanoparticles at both concentrations tested (Fig. 4C). In contrast, CeO2, Al2O3 and CuO nanoparticles showed significant effects on neurotoxicity at 10 µg/ml under our conditions, as determined by a significant decrease in the number of head thrashes, and this trend was conserved at 50 µg/ml (P < 0.0001 for both CeO2concentrations, P = 0.0087 and P < 0.0001 for Al2O3 at 10 µg/ml and 50 µg/ml, respectively and P = 0.0002 for CuO at 10 µg/ml), although the effect on neurotoxicity of CuO nanoparticles at 50 µg/ml was not significant (P > 0.05) (Figs. 4D and 4F).

Figure 4 Neurotoxicity of various nanoparticles (A, Ag; B, SiO2; C, TiO2; D, CeO2; E, Al2O3; and F, CuO) to C. elegans N2 at 0, 10 and 50 µg/ml.

200 L1s were grown in 6-well plates containing S-medium supplemented with described concentrations of each nanoparticle for 72 h at 21 °C. Adult day 1 worms were then washed and put into an unseeded NGM plate containing K-medium and allowed to swim freely for 60 s. The number of head thrashes made by a single worm were then counted for 1 minute. Only three nanoparticles, CeO2, Al2O3 and CuO, show significant differences for the neurotoxicity parameter at 10 µg/ml, compared to control (P < 0.01). Bar-graphs represent average number of head thrashes ±SD per condition.

Based on the phenotyping results, we further performed RNAseq analysis on C. elegans exposed to Ag and SiO2(10 µg/ml) as these two nanoparticles demonstrated the most outstanding effect on majority of the parameters (locomotion velocity, growth and reproduction). For each sample (5 replicates for the Ag, SiO2 and control groups), around 20 million reads were obtained. Approximately 97% of reads were mapped to the worm’s genome and a total of 18,861 gene sequences were identified, using a minimal total read count of 3 across samples. Sample distribution by principal component analysis (PCA) is shown in Fig. S1. Differentially expressed genes (DEGs) based on edgeR between control and the toxicity groups (2,648 DEGs in the Ag group and 1,087 DEGs in the SiO2 group) are shown in Data S1 (FDR < 0.05).

Gene set enrichment analysis (GSEA) based on Gene Ontology biological processes (BPs) showed various statistically enriched positively or negatively based on the fold changes and running enrichment scores (Data S2). The top 20 and phenotype reflecting enriched BPs are shown in Table 2 (FDR < 0.05). These BPs were related to various physiological events such as cellular and metabolic responses. Apoptotic process showed gene enrichment with positive NES in both Ag and SiO2 groups under significant levels (FDR < 0.0001). Phenotype reflecting BPs including regulation of locomotion, reproductive process and cell growth were enriched with negative NES significantly (FDR < 0.0001 for regulation of locomotion, FDR = 0.001 for regulation of reproductive process, FDR = 0.002 for regulation of cell growth in the Ag group; FDR < 0.0001 for regulation of locomotion, FDR = 0.008 for regulation of reproductive process, FDR = 0.042 for regulation of cell growth in the SiO2 group) (Fig. 5). A number of genes were commonly detected within the top five category in the phenotype reflecting enriched BPs for both Ag and SiO2 groups, including transcription factor (che-1) and MiRP K channel accessory subunit (mps-1) in regulation of locomotion, caveolin (cav-1) in regulation of reproductive process, and cyclic nucleotide-gated cation channel (tax-4) and protein let-756 (let-756) in regulation of cell growth (Data S2).

Table 2 Top 20 and phenotype reflecting enriched biological processes.

The enriched biological processes, ranked based on NES and FDR value, are shown. The enriched biological processes, detected by the GSEA analysis, were statistically significant (no asterisk: FDR < 0.0001, *: FDR < 0.05 and **: FDR < 0.01). The phenotype reflecting enriched biological processes are shown under dashed-line ‡: Full name is provided in Data S2.

		Ag				SiO2			
	Biological process		NES	Rank			NES	Rank	
Positive enrichment								
	Cellular process	Cell cycle	6.591	2		Cell cycle	4.095	2	
		Cell cycle process	6.624	1		Cell cycle process	4.167	1	
		Meiotic cell cycle	5.455	10		Meiotic cell cycle	3.413	10	
		Meiotic cell cycle process	4.744	15		_	_	_	
		Meiotic nuclear division	5.215	12		Meiotic nuclear division	3.158	12	
		Meiotic chromosome segregation	4.441	18		_	_	_	
		Mitotic cell cycle	4.805	14		Mitotic cell cycle	2.776	19	
		Regulation of cell cycle	4.378	20		_	_	_	
		Cell division	4.561	17		Cell division	2.886	14	
		Death&cell death	6.022	4		Death&cell death	3.699	7	
		Programmed cell death	5.736	7		Programmed cell death	3.695	8	
		Apoptotic process	6.332	3		Apoptotic process	3.887	5	
		Chromosome segregation	4.943	13		_	_	_	
		Posttranscriptional gene silencing&gene silencing	4.407	19		_	_	_	
	Growth	_	_	_		Developmental growth	2.771	20	
		_	_	_		Regulation of growth	2.787	18	
	Metabolic process	Posttranscriptional regulation of gene gene expression	4.571	16		_	_	_	
	Multicellular organism process	_	_	_		Positive regulation of multicellular organismal process	2.814	16	
	Multi-organism process	Sexual reproduction	5.713	8		Sexual reproduction	3.854	6	
		Gamete generation	5.982	5		Gamete generation	4.012	3	
		Germ cell development&cellular process †	5.334	11		Germ cell development& cellular process †	3.897	4	
		_	_	_		Female gamete generation	2.813	17	
		_	_	_		Oogenesis	3.010	13	
		_	_	_		Spermatid differentiation& spermatid development	2.818	15	
	Cellular component organization or biogenesis	Organelle fission	5.756	6		Organelle fission	3.433	9	
		Nuclear division	5.621	9		Nuclear division	3.364	11	
Negative enrichment								
	Behavior	Single organism behavior	−4.631	1		_	_	_	
	Cellular process	Cell communication	−4.285	7		Aromatic compound biosynthetic process	−5.477	11	
		Cell surface receptor signaling pathway	−4.082	17		Regulation of cellular metabolic process	−5.198	19	
	Developmental process	Cell morphogenesis	−4.571	3		_	_	_	
		Cell projection morphogenesis	−4.576	2		_	_	_	
		Cell part morphogenesis	−4.276	8		_	_	_	
		Generation of neurons	−4.048	20		_	_	_	
									
	Metabolic process								
		_	_	_		Cellular nitrogen compound biosynthetic process	−5.306	16	
		_	_	_		Heterocycle biosynthetic process	−5.423	12	
		_	_	_		Nucleobase containing compound biosynthetic process	−5.486	10	
		_	_	_		Organic cyclic compound biosynthetic process	−5.323	15	
		_	_	_		Positive regulation of gene expression	−5.029	20	
		_	_	_		Regulation of biosynthetic process& †	−5.410	13	
		_	_	_		Regulation of macromolecule biosynthetic process	−5.267	17	
		_	_	_		Regulation of cellular macromolecule biosynthetic process	−5.260	18	
		_	_	_		Regulation of nitrogen compound metabolic process& †	−5.587	9	
		_	_	_		Regulation of primary metabolic process	−5.406	14	
		Regulation of rna metabolic process	−4.138	15		Regulation of rna metabolic process	−5.971	2	
		Rna biosynthetic process	−4.198	12		Rna biosynthetic process	−5.923	3	
		Nucleic acid templated transcription	−4.121	16		Nucleic acid templated transcription	−5.833	6	
		Regulation of rna biosynthetic process	−4.164	14		Regulation of rna biosynthetic process	−5.983	1	
		Transcription dna templated	−4.200	11		Transcription dna templated	−5.864	5	
		Regulation of transcription dna templated& †	−4.062	19		Regulation of transcription dna templated& †	−5.805	7	
		Transcription from rna polymerase ii promoter	−4.268	9		Transcription from rna polymerase ii promoter	−5.881	4	
		Regulation of transcription from rna poly. †	−4.192	13		Regulation of transcription from rna poly. †	−5.775	8	
	Multicellular organism development	Nervous system development	−4.528	4		_	_	_	
	Response to stimulus	Response to external stimulus	−4.493	5		_	_	_	
		Taxis	−4.066	18		_	_	_	
	Signaling	Signaling	−4.266	10		_	_	_	
		Single organism signaling	−4.299	6		_	_	_	
Phenotype reflecting							
	Growth	Regulation of cell growth**	−2.326	167		Regulation of cell growth*	−1.714	410	
	Locomotion	Regulation of locomotion	−3.124	66		Regulation of locomotion	−3.694	86	
	Reproduction	Regulation of reproductive process**	−2.476	132		Regulation of reproductive process**	−2.062	294	

Figure 5 Selected significant GO terms (detected by GSEA).

Enriched BPs including regulation of locomotion, regulation of reproductive process and regulation of cell growth, show negative NES in the Ag group (A–C) and the SiO2 group (D–F) (FDR < 0.05).

The other significantly enriched BPs were associated with various physiological events and immune defense of the organism. As both Ag and SiO2 nanoparticles were foreign substances for the worm, we further examined the expression profiles of genes associated with innate immunity. Compared to Ag and Control, SiO2 exposure led to significant downregulations of genes related to the innate immune response (FDR < 0.0001) (Fig. 6).

Figure 6 Heatmap of differentially expressed genes involved in the innate immune response.

A downregulation pattern is observed in the SiO2 group, in comparison with the Ag group and Control (A). Pink and blue circles in Venn diagram represents differentially expressed genes in the SiO2 and Ag groups, respectively (B). *: Ag, **: SiO2, ***: Common for Ag and SiO2.

The enriched BPs by GOATOOLS were mainly related to the physiological events, of which, some were identical to those observed by the GSEA analysis (Data S2). However, the number of the BPs detected by GOATOOLS was lower compared to those detected by GSEA, which was likely related to the differences in gene inputs (DEGs with FDR < 0.05 vs cut-off free DEGs ranked by fold change) and/or the methodologies used. Nevertheless, the phenotype reflecting BPs including the regulations of locomotion and reproductive process were also enriched in the Ag group in the GOAtools analysis. However, this was not the case for the SiO2 group, which appeared to be related to lower number of DEGs in this group, compared to the Ag group.

Analysis of the RNAseq data using WormExp showed gene enrichment in newly determined terms (Data S2). Some of these terms were found associated with regulations of locomotion, reproduction and cell growth in both groups. The “glp-1 mutant” term was enriched in the Ag and SiO2 groups, which refers to the diminished reproductive capability of C. elegans (Gracida & Eckmann, 2013). In regard to development terms, “pgl-1 mutant” and “P-granule RNAi” were enriched in the Ag and SiO2 groups, respectively, linking regulation of cell growth (Knutson et al., 2017). The “wdr-23 mutant” term was also present for both nanoparticle exposure groups. This is notable as wdr-23, through the action of skn-1, is involved in proper locomotion of C. elegans (Staab et al., 2013). Finally, the other terms, including regulation by heavy metals/NPs (such as Ag), were found for both groups, further validating the nanoparticle effect. Altogether, the WormExp gene enrichment terms, obtained from the RNAseq data, appear to be in agreement with the phenotypic assay results.

Pathway enrichment analysis against the KEGG database showed various significantly enriched pathways (Data S3). The top 20 enriched biological pathways are shown for both exposures in Table 3. Ribosome, proteasome, aminoacyl-tRNA biosynthesis and RNA transport were significantly upregulated in both groups, indicating overall higher rate of protein turnover upon exposure. In contrast, biological pathways reflecting phenotypes including neuroactive ligand–receptor interaction [regulating locomotion (Kong et al., 2015)], wnt-signaling (regulating reproduction and cell growth (Inoki et al., 2006; Hernandez Gifford, 2015) and MAPK signaling (regulating reproduction and cell growth (Zhang & Liu, 2002; Andrade et al., 2014) were significantly down-regulated in both exposures (Fig. 7). Some genes were found commonly within the top 5 enriched gene category in each enriched biological pathway for both Ag and SiO2 treatments (based on fold change and running enrichment score), which were tachykinin receptor family (tkr-2), serotonin/octopamine receptor family (ser-1) in neuroactive ligand–receptor interaction pathway, skp1 related (ubiquitin ligase complex component, skr-8, skr-10, skr-12) in wnt signaling pathway, protein ver-1 (ver-1) and heat shock protein (hsp 70) in MAPK signaling pathway (Data S3).

Table 3 Top 20 enriched biological pathways.

Ranking of the enriched biological pathways is based on NES and FDR value. The enriched biological pathways (KEGG) by the enrichment analysis were found under statistically significant levels (FDR ¡ 0.05) except those indicated with asterisk (*: 0.05 < P < 0.13; 0.12 < FDR < 0.21).

	Ag			SiO2			
	Biological pathway	NES	Rank	Biological pathway	NES	Rank	
Positive enrichment	Ribosome	7.179	1	Ribosome	6.864	1	
	Proteasome	4.731	2	Proteasome	3.682	2	
	Rna transport	4.484	3	Aminoacyl-trna biosynthesis	3.296	3	
	Spliceosome	4.453	4	Oxidative phosphorylation	2.986	4	
	Oxidative phosphorylation	4.320	5	Carbon metabolism	2.969	5	
	Aminoacyl-trna biosynthesis	4.078	6	Rna transport	2.955	6	
	Ribosome biogenesis in eukaryotes	3.661	7	Pyruvate metabolism	2.724	7	
	Nucleotide excision repair	3.643	8	Rna polymerase	2.439	8	
	Carbon metabolism	3.458	9	Fanconi anemia pathway	2.337	9	
	Glycosylphosphatidylinositol (gpi)-anchor biosynthesis	3.349	10	Ribosome biogenesis in eukaryotes	2.333	10	
	Fanconi anemia pathway	3.330	11	Fatty acid metabolism	2.280	11	
	Dna replication	3.316	12	Nucleotide excision repair	2.170	12	
	Pyrimidine metabolism	3.141	13	Pyrimidine metabolism	2.169	13	
	Rna polymerase	3.021	14	Fatty acid degradation	2.160	14	
	Mrna surveillance pathway	2.926	15	Rna degradation	2.157	15	
	Endocytosis	2.840	16	Valine, leucine and isoleucine degradation	2.144	16	
	Rna degradation	2.816	17	Biosynthesis of amino acids	2.140	17	
	Mismatch repair	2.762	18	Glycosylphosphatidylinositol (gpi)-anchor biosynthesis	2.087	18	
	Homologous recombination	2.747	19	Dna replication	2.066	19	
	Peroxisome	2.699	20	Glycolysis / gluconeogenesis	2.065	20	
Negative enrichment							
	Neuroactive ligand–receptor interaction	−2.490	1	Protein processing in endoplasmic reticulum	−3.896	1	
	Wnt signaling pathway	−2.350	2	Endocytosis	−3.374	2	
	Lysosome	−2.199	3	Spliceosome	−3.298	3	
	Ecm-receptor interaction	−2.180	4	Wnt signaling pathway	−3.199	4	
	Phagosome	−2.118	5	Ubiquitin mediated proteolysis	−2.849	5	
	Mapk signaling pathway	−1.973	6	Tgf-beta signaling pathway	−2.782	6	
	Calcium signaling pathway	−1.921	7	Mrna surveillance pathway	−2.738	7	
	Drug metabolism - cytochrome p450	−1.872	8	Mapk signaling pathway	−2.600	8	
	Autophagy - animal	−1.726	9	Calcium signaling pathway	−2.248	9	
	Age-rage signaling pathway in diabetic complications	−1.713	10	Ecm-receptor interaction	−2.077	10	
	Tgf-beta signaling pathway	−1.667	11	Phosphatidylinositol signaling system	−2.068	11	
	Glycosphingolipid biosynthesis - globo and isoglobo series	−1.593	12	Notch signaling pathway	−1.951	12	
	Erbb signaling pathway	−1.590	13	Autophagy - other	−1.946	13	
	Polyketide sugar unit biosynthesis	−1.589	14	Hippo signaling pathway -multiple species	−1.913	14	
	Taurine and hypotaurine metabolism*	−1.601	15	Autophagy - animal	−1.866	15	
	Glycosaminoglycan degradation*	−1.578	16	Inositol phosphate metabolism	−1.839	16	
	Autophagy - other*	−1.450	17	Neuroactive ligand–receptor interaction	−1.828	17	
	Hippo signaling pathway -multiple species*	−1.433	18	Phagosome	−1.816	18	
	Retinol metabolism*	−1.413	19	Mitophagy - animal	−1.804	19	
	Mitophagy - animal*	−1.376	20	Basal transcription factors	−1.700	20	

Discussion

As the use of nanoparticles has increased dramatically in recent years, there is a growing concern regarding their potential impact to environment and human health. In this study, we have systematically evaluated a C. elegans-based animal model for nanotoxicity assessment. Our results have shown that Ag and SiO2 have the most potent toxic effect on locomotion velocity and growth, as well as reproduction (brood size), but not on neurotoxicity. In this model, the transcriptome profile is concordant with the phenotype characteristic for both exposures (Fig. 8).

The top 20 GO BPs identified by the GSEA were related to various physiological events in the Ag and SiO2 toxicities. The exposure to both nanoparticles downregulated multiple regulatory biological processes, including regulation of locomotion, regulation of reproduction and regulation of cell growth, which was consistent with the phenotype profiling of our study.

Dysfunction of the enriched genes including transcription factor che-1 (che-1) and MiRP K channel accessory subunit (mps-1) (regulation of locomotion), caveolin (cav-1) (regulation of reproductive process), and cyclic nucleotide-gated cation channel (tax-4) and protein let-756 (let-756) (regulation of cell growth), have been previously shown to hinder the worm’s biological events (Uchida et al., 2003; Bianchi et al., 2003; Scheel et al., 1999; Komatsu et al., 1996; Roubin et al., 1999). Inactivation of che-1 (mediating chemotaxis through ASE neurons) and of mps-1 (a voltage-gated pore-forming potassium subunit) impairs neuronal activities such as chemotaxis and locomotion (Uchida et al., 2003; Bianchi et al., 2003). Caveolin-1 (cav-1), identified in the adult germ line and highly expressed in eggs, is required for Ras/MAP-kinase-dependent progression (Scheel et al., 1999). Inhibition of tax-4 and let-756 genes hinders chemosensation and results in larval arrest, respectively (Komatsu et al., 1996; Roubin et al., 1999).

Figure 7 Selected significant KEGG pathways (detected by GSEA).

Neuroactive ligand-receptor interaction, wnt signaling pathway and MAPK signaling pathway, negatively enriched based on NES, are shown in the Ag group (A–C) and the SiO2 group (D–F) (FDR < 0.05).

Figure 8 Proposed phenotype and transcriptome relationship in Ag and SiO2 toxicities.

The gene enrichment profiles of the biological processes and pathways are concordant with phenotype characteristics for both toxicities. Arrows indicate gene enrichment profiles with negative or positive manner.

Although biological processes reflecting phenotypes were similar between both toxicities in our study, SiO2 nanoparticles induced a remarkable downregulation pattern in innate immune response, compared to Ag and Control. In particular, several C-type lectins, which are known to be important components in innate immunity (Mayer, Raulf & Lepenies, 2017), appeared to be exclusively downregulated by SiO2 nanoparticles. The subject of nanoparticle exposure and the effects on immune system has been an active research area (Boraschi et al., 2017). Exposure to nanoparticles has been linked to changes in the immune response such as inflammation, hypersensitivity and immunosuppression and has been shown to induce such responses through antigen-presenting cells in humans, highlighting the interaction between nanoparticles and the innate immune response (Alsaleh & Brown, 2018). Biocoronas, formed by the interaction of the nanoparticle surface with proteins and lipids, are highly reactive immunologically and have recently gained the attention of regulatory agencies (Shannahan, 2017).

The top 20 significant KEGG pathways identified in both exposure studies are similar to the findings based on GO BPs. In particular, the regulatory biological pathways linked to phenotypes, including neuroactive ligand–receptor interaction (relates to locomotion Inoki et al., 2016), wnt and MAPK signaling pathways (relates to reproduction and cell growth (Inoki et al., 2016; Hernandez Gifford, 2015; Zhang & Liu, 2002; Andrade et al., 2014)), were found to be downregulated in both experiments. The pathway interaction analysis by ClueGO showed that wnt signaling pathway was interacting with tgf-beta pathway which was also enriched with significantly negative NES in both toxicities. These signaling pathways are known to interact with each other and control adult tissue homeostasis (Warner, Greene & Pisano, 2005). The downregulation of genes involved in neuroactive ligand receptor interaction likely to be responsible for the changes in locomotion.

The genes within the top 5 enriched gene category in the indicated pathways, commonly observed in both toxicities, were tachykinin receptor family (tkr-2), serotonin/octopamine receptor family (ser-1) (neuroactive ligand–receptor interaction pathway), skp1 related (ubiquitin ligase complex component, skr-8, skr-10, skr-12) (wnt signaling pathway), and protein ver-1 (ver-1) and heat shock protein (hsp 70) (MAPK signaling pathway). The proteins encoded by tachykinin receptor and ser-1 genes regulate locomotion via mediate neurotransmission and indirect modulation of neuromuscular circuits, respectively (Pennefather, 2004; Dernovici et al., 2007). The proteins encoded by skp1 related genes (such as skr-8 and skr-10) are known to be a core element of SCF ubiquitin-ligase complexes and involved in posterior body morphogenesis, embryonic and larval development, and cell proliferation in C. elegans (Nayak et al., 2002). Putative vascular endothelial growth factor receptors (VERs) of C. elegans and Hsp70 chaperones act in the PVF-1 signalling pathway for ray 1 positioning (in the male worms) and mediate protein folding, influencing various regulatory proteins, respectively (Dalpe et al., 2013; Mayer & Bukau, 2005). Overall, these genes may play significant roles on locomotion, reproduction and cell growth in response to nanotoxicity.

Apart from the many similarities observed between the two nanoparticle effects, Ag and SiO2 also showed opposite effects on some of the top 20 enriched biological pathways. Spliceosome, mRNA surveillance and endocytosis pathways were enriched with positive NES in Ag, but with negative NES in SiO2, despite no obvious phenotypic differences were observed in our studies.

Comparison of the transcriptomics changes during Ag and SiO2 exposures with the findings in previous studies on metal toxicities indicated that metallothionein-2 (mtl-2), a commonly observed responsive gene to the metal toxicities (conserved in C. elegans and mammals) (Caito et al., 2012; Cui et al., 2007; Roh, Lee & Choi, 2006; Kumar et al., 2015; Anbalagan et al., 2012), was up-regulated in these groups (FDR =0.002 for Ag, FDR = 0.043 for SiO2), which further confirmed the effectiveness of the model.

Ag and SiO2nanoparticles have been shown to affect locomotion velocity in C. elegans N2, as previously described (Jung et al., 2015). In addition, Ag and SiO2 have been shown to reduce brood size, according to several studies (Kleiven et al., 2018; Wu et al., 2013; Pluskota et al., 2009). In contrast, we did not observe a significant decrease in brood size to exposure to 10 and 50 µg/ml of TiO2 nanoparticles, as observed by Wu et al. (2013), indicating differences in our conditions, as these authors saw a slight decrease in progeny production when using these nanoparticles in the µg/l concentration range. The effects of the other nanoparticles used in this study (i.e., CuO, Al2O3, and CeO2) are relatively unknown based on the literature. In terms of neurotoxicity experiments, as determined by the number of head thrashes per minute in our study, our results differ from published results (Wu et al., 2013; Pluskota et al., 2009; Piechulek & Von Mikecz, 2018) which show that Ag and SiO2 are neurotoxic at lower concentrations than the ones used in this study, although SiO2 nanoparticles were found to have a modest effect at 50 µg/ml on neurotoxicity under our conditions. Indeed, we noticed that many neuron system-related BPs or pathways were enriched with negative NES in both Ag and SiO2 groups appear to have limited influence on head thrashing in our analysis. Additionally, our study is different compared to the indicated studies, as we only looked at the effect of nanoparticles incubated from the L1 stage to the adult day one stage. We observed that CeO2, Al2O3 and CuO nanoparticles had significant neurotoxicity at 10 µg/ml, as determined by the decrease in number of head thrashes per minute, which is a novel observation. The reported differences in this study may be because the Ag and SiO2 nanoparticles used in our study were confirmed to have a spherical shape under the manufacturer’s test conditions (Table 1), whereas the others (TiO2, CeO2, Al2O3 and CuO) were unconfirmed to adopt any shape at all. Further study is required to elucidate the answer to this question. It was shown that co-feeding nanoparticles with E. coli OP50 in S-medium leads to the uptake of these nanoparticles through the pharynx and absorption through the gut (Piechulek, Berwanger & Von Mikecz, 2019). We speculate this to be true as well under our conditions. Silica (SiO2) nanoparticles were shown to inhibit the peptide transporter OPT-2/PEP-2, present on the apical layer of the intestinal membrane in C. elegans (Piechulek, Berwanger & Von Mikecz, 2019). Inhibition of the OPT-2/PEP-2 transporter leads to the accumulation of silica nanoparticles in gut granules, indicating they are taken up within the organism. Fang-Yen et al. showed that particles with a diameter range of 0.5 µm to 3 µm are taken up by the pharynx (Fang-Yen, Avery & Samuel, 2009). We propose that this size range is circumvented when nanoparticles are co-fed with E. coli OP50 to gain access to the gut for absorption.

The observed effects in the various experiments can be either nanoparticle-specific or compound-/element-specific. Nanoparticles, ranging from 1 to 100 nm in size, are larger than their elemental constituents, which are metal cations under our experimental conditions. We reason that the observed effects in the various experiments and RNAseq analysis are due mainly to nanoparticles, although nanoparticles, such as Ag, release positively-charged ions upon incubation in liquid media and the proportion of released cations is small (Lekamge et al., 2018). In addition, metal oxide nanoparticles tend to release ions into liquid medium depending on the cationic metal charge (Simeone & Costa, 2019). For example, where Z is the oxidation number of the metal cation, oxides of nanoparticle cations with Z ≤ 2 dissolve more than 10%, whereas this fraction is reduced to less than 1% for nanoparticle oxides of cations with Z >3 (Simeone & Costa, 2019). According to this relationship, we would expect that less than 1% of the metal oxide nanoparticles used in our study to be dissolved, except for CuO nanoparticles which has a metal cation Z value of 2. As evidence for our reasoning, the WormExp enrichment analysis identified the “Ag NPs” term (Data S2).

To address whether the nanoparticles used in our study affected E. coli OP50 growth and thus C. elegans feeding, we incubated E. coli OP50 with 50 µg/ml of each nanoparticle in S-medium (Fig. S2). Compared to E. coli OP50 alone, the treated E. coli OP50, with the highest concentration of nanoparticles (50 µg/ml), demonstrated only slight growth defects, as determined by measuring bacterial density (Fig. S2). Antimicrobial nanoparticles, such as Ag nanoparticles, are antimicrobial as they interact with bacterial membranes and proteins through the released metal cations (Sondi & Salopek-Sondi, 2004). However, this effect seems to be minimal under our experimental conditions (Fig. S2), likely due to the low dissolution rate of Ag+ cations (Lekamge et al., 2018). Ag nanoparticles had only a major effect on bacterial density after five days compared to the other nanoparticles tested (Fig. S2). The same was observed for CuO nanoparticles (Fig. S2). Therefore, a constant source of E. coli OP50 food, as determined by bacterial density, was available during the course of the various experiments.

Conclusions

The aim of this study was to evaluate the effects of various nanoparticles on C. elegans using standard phenotyping assays and characterize transcriptomics changes of the worms exposed to selected nanoparticles (which showed the toxic effects for the majority of the parameters tested). With all these observations, we provide a novel angle to study the toxicity of nanoparticles on organisms, by exploring the mode of action of Ag and five metal oxide nanoparticles on different life history endpoints in C. elegans. To the best of our knowledge, this is the first study that integrates phenotype screening with RNAseq to investigate nanotoxicity in intact animals using C. elegans. Our RNAseq data not only confirmed positive enrichment of apoptotic process as reported in the literature (McShan, Ray & Yu, 2014; Clement & Jarrett, 1994; Kim et al., 2015), it also revealed that toxicities induced by both nanoparticles have down-regulated genes in multiple important regulatory biological processes and pathways, with opposite effects on innate immunity.

Supplemental Information

Figure S1 PCA analysis

Sample distributionfor each toxicity group (in comparison with control) is shown.

Click here for additional data file.

Figure S2 Effect of tested nanoparticles on E. coli OP50

E. coli OP50 was grown in 50 µg/ml of CeO2, Ag, SiO2, TiO2, Al2O3 and CuO nanoparticles in S-medium in a 12-well plate at 21 °C. Bacterial density was measured by taking the OD595 nm values of three independent wells. Statistical significance, after five days incubation, is indicated with an asterisk. * P < 0.05, *** P < 0.01, **** P < 0.0001, compared to E. coli OP50 alone.

Click here for additional data file.

Data S1 Differentially expressed genes by edgeR in Group Ag and Group SiO2

A total of 2648 genes were differentially expressed in Group Ag and a total of 1087 genes were differentially expressed in Group SiO2, in comparison with Control (FDR < 0.05).

Click here for additional data file.

Data S2 Gene ontology enrichment by GSEA and GOATOOLS and newly determined term enrichment by WormExp

Enriched biological processes showing up- and down-regulation patterns are shown for both Ag and SiO2 groups (P < 0.05). Genes with description for the top 20 enriched biological processes are provided.

Click here for additional data file.

Data S3 KEGG pathway enrichment by GSEA

Enriched biological pathways showing up- and down-regulation patterns are shown for the Ag and SiO2 groups (P < 0.05). Genes with description for the top 20 enriched biological pathways are provided.

Click here for additional data file.

Supplemental Information 1 Raw data for Fig. S2

E. coli OP50 growth, measured at OD 595 nm, is shown with tested nanoparticles.

Click here for additional data file.

Supplemental Information 2 Raw data/raw numbers for Figs. 1–4

Raw data/raw numbers for locomotion velocity, body length, reproduction capacity and neurotoxicity for Figs. 1–4 are provided, respectively.

Click here for additional data file.

Additional Information and Declarations

Competing Interests

Author Contributions

Data Availability

Jianguo Xia is an Academic Editor for PeerJ.

Charles Viau and Orçun Haçariz conceived and designed the experiments, performed the experiments, analyzed the data, prepared figures and/or tables, authored or reviewed drafts of the paper, and approved the final draft.

Farial Karimian performed the experiments, prepared figures and/or tables, and approved the final draft.

Jianguo Xia conceived and designed the experiments, authored or reviewed drafts of the paper, and approved the final draft.

The following information was supplied regarding data availability:

The raw data for Figs. 1–4 and Fig. S2 are available in the Supplemental Files. The RNAseq dataset is available in the NCBI Gene Expression Omnibus (GEO) database: GSE122728.

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
