# Peer review of "Comprehensive phenotyping and transcriptome profiling to study nanotoxicity in C. elegans"

_PeerJ, doi:10.7717/peerj.8684_

## Round 0.1 · original submission · Major Revisions

As you can see below, the Reviewers had overall positive opinions about your manuscript. However, they had some suggestions and concerns which I am kindly asking you to address prior to further considering the manuscript for publication.

·

Basic reporting

The manuscript had many typos and instances where words were entirely missing. One highlight is "in contract" on line 258. I recommend the author go over the manuscript again with basic grammar and syntax in mind.

There was nice background and context provided.

One issue I had was with some acronyms such as NES. Which was mentioned multiple times in the results section without an explanation of the acronym. I might have missed the declaration of it before.

Experimental design

I thought the experimental design was well planned and seemingly well executed. The authors presented novel research that has relevance for basic biology as well as relevance in with regard to translational medicine in humans. I thought the methods were well explained and I was not at a loss when considering the motivation and execution of the experiments.

I think that some tools and annotations used by the author are out of date such as GSEA since they are relying so heavily on the gene set enrichment analysis of the project it would be nice to use an additional tool (Such as GOAtools https://www.nature.com/articles/s41598-018-28948-z) to see if the see the same results.

The wbcel235 annotation is also quite old and the authors would find potentially more genes of interest if they use the current wormbase version (WS271).

Validity of the findings

I do not doubt the validity of the findings.

I think that the amount of differential expression (DE) of genes found in the two RNA-seq experiments should be put in the paper and not put in a supplemental. The amount of DE genes found has a high impact on which biological pathways are reported.

In addition, it is easy to copy and past 20 enriched biological pathways but it is hard to make sense of the results. I commend the authors for doing their homework and interpreting their data but with so many pathways it is easy to be speculative about the validity of their findings. I think RT-PCR validation on the important genes mentioned by the authors would do well to boost the RNA-seq data.

Reviewer 2 ·

Basic reporting

Basic reporting:
In general manuscript is well written with a few changes suggested.
Line 1: Remove “of”
Line 32: Consider rephrasing as not clear what “the nanotechnology” is defined to include.
Also imprecise language is often used such as around, or ~ and this weakens the impact of the manuscript. This will be explored further in the experimental design section also.
Line 165 & 166: Unnecessary repetition from methods.

Experimental design

This research matches the aims and scope of this journal.

Research question is not very well defined and this is reflected in the experimental design.
Whilst the methods are well described and could be replicated, the overall research question is very general and chosen methods do not follow a clear line of questioning and argument despite still providing some useful data.

Validity of the findings

The major issue is centred on questions of uptake of these nanoparticles. In table 1. Particle size is shown to vary between each treatment. Previous literature has demonstrated the importance of particle size and uptake of particles in C. elegans (eg. Fang-Yen, C., Avery, L., & Samuel, A. D. (2009). Two size-selective mechanisms specifically trap bacteria-sized food particles in Caenorhabditis elegans. PNAS, Flavel, M., et al. (2018) "Growth of Caenorhabditis elegans in defined media is dependent on presence of particulate matter." G3: Genes, Genomes, Genetics. The variation of size introduces a variable to the experiment that needs to be addressed by the authors. Furthermore, if there is a variation to the uptake the relative concentrations into the worm could vary significantly and concentration of these particles and compounds inside the worm should be measured.
Addressing the variation of size in these particles would also include at least some speculation or ideally experimental data on how these particles are being taken up. At 100 micron, these particles are not far outside the edible/ bacterial size range for C. elegans. Therefore, even though steps have been taken to prevent aggregation, aggregates of as little as 2-5 particles would be likely to be ingested at a similar size to food and therefore the mode of uptake is no longer analogous to toxicity in larger animals such as humans. If these particles are being ingested as food this study becomes less about the effect of nanoparticles and more about the toxicity of ingesting silver, titanium etc which reduces the novelty of this work significantly and also as already discussed the concentration of these compounds inside the worm has not been controlled for. If the uptake is likely to be across the cuticle some imaging would be helpful to see where these particles accumulate and if they are able to be excreted. Overall, it is also unclear from the evidence presented whether the relative size difference between a worm and these particles is different enough to justify the claim that this is a useful tool for profiling nanotoxicity, if the ultimate application of this research is to compare the toxicity effect of these same sized particles with other much larger animals in the environment (fish, humans etc) .
The secondary issue is that whilst the RNAseq data is useful other assays both phenotypic and more general could have been conducted to align with this data. Other general phenotypic measures such as lifespan, response to stress and behavioural assays would strengthen this manuscript. A dose response to these particles would have also been useful to understand the effect or potential benefit of reducing these nanoparticles in the environment.
Line 87 & 91: Numbers of worms used for each assay varies and is not reported accurately. For experiments of this size it is not clear why a set number of worms was used and reported. Especially as assays are short and worms are unlikely to be censored. If they have been censored this does not appear in raw data or discussed in text.
Line 108, 119: Unclear why more precise worm numbers were not used

Additional comments

I commend the authors for preparing this manuscript. The data provided is a good indicator of some relevant toxicity parameters. The use of advanced techniques, particularly the RNAseq strengthens the manuscript and provides novel data that would be interesting to both the C. elegans and wider research community. However, there are currently some major issues with the manuscript that would need addressing, before it could be accepted for publishing in PeerJ.

---

## Round 0.2 · Minor Revisions

Please see the additional comments made by Reviewer 2 and consider adding discussion on the topics outlined, i.e., usefulness of C. elegans as a model for nanotoxicology; explanation if the detected effects in C. elegans might have been nanoparticle-specific or compound/element-specific; and if nanoparticle exposures may have affected the viability of E. coli and thus, feeding of C. elegans. I believe discussing these aspects would further enhance the significance of the results and the clarity of the paper to the reader.

Reviewer 2 ·

Basic reporting

The authors have made all necessary improvements to basic reporting required.

Experimental design

Experimental design has not changed since last revision.

Validity of the findings

I accept the arguments put forward by the authors and agree it is valid to infer that uptake of nanoparticles has occured. The lack of control or measurement of this process still causes me concerns.

The authors speculate in their rebuttle that there is unlikely to be uptake across the cuticle.This again raises the question of how reliable a model for nanoparticle toxicity this presents, if this critical uptake pathway for nanoparticles in higher animals is not present in this model. I accept that there is disease and cellular homology that makes C. elegans a useful model for general toxicology, but the authors should include more discussion of what makes C. elegans a useful model for nano-particle focused toxicology.

If uptake is entirely through the pharynx, then clearer comparison to other studies where these same elements and comparable concentrations have have been conducted on C. elegans, but not using nanoparticles is required. It is unclear currently whether the toxic effects are related in any way to the delivery method of the compounds being through nanoparticles or if it is only a consequence of an increased dietary intake of silver and silica. Because the article is framed around investigating nanoparticle toxicity, not a toxicity experiment using traditional methods and therefore this distinction needs to be fully explored to meet the aim of the study.

If the uptake is mediated by co-feeding with OP50 E. coli, is there any effect on the viability of E. coli? Whilst uptake may have occurred it is also unclear still if there is evidence that this uptake is constant between treatment groups.

Additional comments

Most of the feedback from the first submission of this manuscript has been addressed. However, there are some remaining issues with the manuscripts fundamental aim that need to be addressed. These are described in detail in the validity of findings section. Overall, I accept the findings are of interest to the C. elegans field, due largely to the RNAseq data. However, I do not recommend publication unless the short comings of this experiment are overcome, or at least thoroughly justified in the manuscript text.

---

## Round 0.3 · accepted · Accept

I have no further comments.